**Data Availability Statement:** No datasets were generated or analysed during the current study. All

# FoRSHE-X digital health intervention to improve the quality of life during chemotherapy among gynecological cancer survivors in Indonesia: A protocol for a pilot and feasibility study

Yati Afiyanti[1☯], Dyah Juliastuti[2☯]*, Winnie Kwok Wei So[3‡], Ariesta Milanti[4‡], Lina Anisa Nasution[1,5‡], Aprilia Dian Prawesti[1‡]

**1** Faculty of Nursing, Universitas Indonesia, Depok, Indonesia, **2** Program Study of Nursing, Faculty of Health Sciences, Universitas Ichsan Satya, South Tangerang, Indonesia, **3** The Nethersole School of Nursing, The Chinese University of Hong Kong, Hong Kong, **4** Faculty of Nursing and Midwifery, Binawan University, Jakarta, Indonesia, **5** Program Study of Nursing, Faculty of Sport and Health Education, Universitas Pendidikan Indonesia, Bandung, Indonesia

☯ These authors contributed equally to this work.
‡ WKWS, AM, LAN and ADP also contributed equally to this work.
* dyahjuliastuti2@gmail.com

## Abstract

Most Indonesian gynecological cancer survivors have unmet supportive care needs during chemotherapy, which may lower their quality of life and discontinue the treatment. Digital health intervention can address this issue. This pilot investigation aims to (1) examine the feasibility and acceptability of a Fighting on distRess, Self-efficacy, Health Effects, and seXual issues (FoRSHE-X) intervention and (2) evaluate prospectively the impact of the study implementation on the level of distress, self-efficacy, side effects' knowledge and management, and sexual quality of life using the RE-AIM (Reach Effectiveness, Adoption, Implementation, and Maintenance) framework. This is a non-randomized mixed-methods pilot study. We will recruit women diagnosed with gynecological cancer undergoing chemotherapy to participate in the FoRSHE-X intervention consisting of ten weeks of social media-based education and telecoaching. We will evaluate the primary outcomes of study feasibility and acceptability, and the secondary outcomes of study impacts at three time points with quantitative and qualitative inquiries. We anticipate a minimum of 30 participants to enroll in the study and complete the assessment. We will disseminate results through conferences and peer-reviewed scientific journals. This study will imply whether a definitive trial to evaluate the potential benefits of the FoRSHE-X is viable and how it should proceed. The protocol can aid researchers or nurses in implementing this approach in their study or practice.

**Trial registration: Clinical trial registration number:** ISRCTN13311651.

relevant data from this study will be made available upon study completion.

**Funding:** This work was supported by the Hibah Publikasi Terindeks International (PUTI) Q1 from the Direktorat Riset dan Pengembangan, Universitas Indonesia (Grant number: NKB 343/UN2.RST/HKP.05.00/2023). The funders had no role in study design, data collection and analysis, decision to publish, or preparation of the manuscript.

**Competing interests:** The authors have declared that no competing interests exist.

# Introduction

Cancer is the second leading cause of mortality in the world, with 10 million cancer-related deaths estimated to occur in 2020 [1]. According to the World Health Organization (WHO), low- and middle-income countries (LMICs) bear the most tremendous burden of cancer [2]. More than 70% of the 10 million cancer-related fatalities were in LMICs [2]. In the case of cervical cancer, the disparity is even more prominent, as 90% of new cases and deaths happened in LMICs [1]. Other gynecological cancer types, including ovarian and uterine corpus cancer, also contribute significantly to the burden of cancer in LMICs [1].

Given the immense cancer concerns in many LMICs, reducing premature mortality through promoting well-being has been listed as one of the targets (target 3.4) of the Sustainable Development Goals (SDG) [3]. Therefore, increasing the country's capacity to provide quality care across the cancer care continuum, including survivorship care, is necessary. Although cancer has been set among the top national catastrophic diseases to prioritize in Indonesia, survivorship care is still suboptimal [4].

An individual can be considered a cancer survivor at the time of diagnosis through the remain of his or her life [5]. Earlier studies showed that most Indonesian gynecological cancer survivors had unmet needs, especially regarding information and comprehensive care [6, 7]. The survivors reported that they hardly had enough time and opportunity to discuss with the health providers, particularly the nurses, the physical, psychosocial, sexual, and other issues that arose during and after cancer therapy [6–8]. The availability of cancer survivorship educational media in Bahasa Indonesia is low, and most Indonesian cancer survivors also have inadequate health literacy [9]. These conditions might hamper the continuity of care and lead to an unanticipated relapse [10]. In addition, unmet needs were also found to be linked to the low quality of life of gynecological cancer survivors in Indonesia [11].

In the digital era, many digital devices and platforms, including mobile phones, mobile applications, websites, and wearable devices, are increasingly used in healthcare services [12]. Digital technologies can be harnessed to enhance communication between cancer patients and healthcare providers [13]. Indeed, prior systematic reviews showed promising findings of the digital interventions' effects in addressing cancer survivors' unmet health information needs and physical and psychosocial support [14–17]. The interventions were delivered through, for instance, social media, messaging, or video-conferencing to connect cancer patients with their health providers, enhance their positive health behaviors, and complete their treatment. Digital interventions can serve as an excellent means to assist cancer survivors in managing their disease and treatment side effects, thereby improving their quality of life and the continuity of the therapy [14, 17, 18]. Digital health interventions were also found to be as effective as usual care to improve the quality of life among cancer survivors [18]. Given their broad potential benefits, the WHO has recommended using digital technologies to strengthen the national health system and achieve the vision of health for all [12].

Nursing intervention for gynecological cancer survivors has not yet been well developed to provide comprehensive and culturally sensitive services in Indonesia, incorporating new technology. Gynecological cancer survivors undergoing cancer therapy will have a better prognosis if they can complete their treatment and understand health management during chemotherapy. Throughout therapy, gynecological cancer survivors have multidimensional needs that, if not well anticipated, may result in therapy termination, relapse, or even death [19]. These needs encompass prevention of recurrence, adopting a healthy lifestyle, managing physical symptoms arising from cancer and its therapy, managing psychosocial and sexual issues, and obtaining information related to self-care [20, 21].

We are developing a sustainable intervention that embraces digital technologies to overcome the communication barriers between cancer survivors and healthcare providers and to

improve self-management for the physical, mental, social, and sexual effects of chemotherapy in the patient's life. We aim to leverage a digital health intervention to address the insufficiency of survivorship care for Indonesian gynecological cancer survivors by combining social media-based education and telecoaching. Therefore, we propose a digital solution called Fighting on distRess, Self-efficacy, Health Effects, and seXual issues (FoRSHE-X).

We will conduct a study to examine the feasibility of the FoRSHE-X intervention using the RE-AIM (Reach, Effectiveness, Adoption, Implementation, and Maintenance) Framework with the specific primary objectives as follows (1) to examine the eligibility and acceptability of the participants towards the study intervention, (2) to evaluate the feasibility of digital education, telecoaching, and intervention facilitators, (3) to evaluate the process of the implementation, and (4) to evaluate the intervention sustainability or participants' involvement over time. Additionally, the secondary objective is to evaluate the impact of the pilot study on the level of anxiety, self-efficacy, side effects' knowledge and management, and sexual quality of life and the participants' perspectives towards this study intervention.

## Methods

### Design plan

We will conduct a non-randomized mixed-methods pilot and feasibility study in which all participants will receive the FoRSHE-X intervention. We will adapt the RE-AIM Framework of O'Brien and colleagues [22] to evaluate the feasibility and pilot study (Table 1). The RE-AIM dimensions represent multiple tasks of complex multi-component intervention [22]. This feasibility study asks questions: Can a definitive trial to assess the effect of an intervention be conducted? Should we proceed with it, and how? [23]. A pilot study is a subset of a feasibility

**Table 1. Defining objectives and data collection using RE-AIM framework.**

| RE-AIM Dimension | Objectives | Quantitative Data | Qualitative Data |
|---|---|---|---|
| **Reach** | To examine the eligibility and acceptability of the participants. | Demographic and clinical data | Participants' experience and perspective on the acceptability of the recruitment process and involvement in this study. |
| | | Acceptance of the research invitation | Facilitators' concerns and challenges in recruiting potential participants. |
| **Effectiveness** | To evaluate the impact of the intervention outcomes. | Chemotherapy side effects | Participant's experience and perspectives on the effect of FoRSHE-X intervention on their health. |
| | | Self-efficacy | |
| | | Distress Thermometer | |
| | | SQoL-F | |
| **Adoption** | To evaluate the feasibility of digital education, telecoaching, and intervention agents. | Complete response to the questionnaire | Researcher daily memoing |
| | | Facilitators satisfaction | Facilitators' experience with digital education and telecoaching. |
| **Implementation** | To evaluate the process of implementation (usability). | Coaching log book | Participants desire and challenges to follow the intervention. |
| | | Proof of telecoaching attendance | |
| **Maintenance** | To evaluate the intervention sustainability or participants' involvement over time. | Participant's diary notes | Participant's experience and perspectives towards digital education and telecoaching activities. |
| | | Participant's adherence on telecoaching | |
| | | Follow-up rate | |
| | | Participant retention | |

study in which the future trial is entirely or partially performed on a smaller scale [23]. Quantitative and qualitative approaches will be used to achieve the study objectives.

## Study settings

The study will be initiated at the Dharmais National Cancer Center (DNCC) in Jakarta, Indonesia. It is a comprehensive cancer center providing cancer care, education, research, and data in Indonesia. As a national referral hospital, it serves as the central hub for cancer diagnosis, treatment, and management, serving patients from Jakarta and nationwide. DNCC offers chemotherapy in several units, including the Systemic Therapy Unit, One Day Care Unit, and Inpatient Unit, in which the potential participants will be recruited, informed about the study, and signed the research consent. Then, the overall study activities will take place online. Digital assessment forms and educational materials are prepared to be accessed by the participants from their homes.

The core team for this research consists of nurses with expertise in cancer care. The principal investigator is a professor of nursing who will oversee the study and deliver the telecoaching intervention. The research team members include nurses working in academia and at the DNCC. The nurses at DNCC will help provide a list of patients who may be eligible for this study. In addition, a medical oncology consultant will also be involved in the study implementation process.

The protocol of this study has been approved by the Ethics Committee of Nursing Faculty, Universitas Indonesia (Approval number: KET-136/UN2.F12.D1.2.1/PPM.00.02/2023) on 29th May 2023 and the committee of the Medical Research Ethics of the Dharmais Cancer Hospital (Approval number: 259/KEPK/VIII/2023) on 8th August 2023. The ISRCTN registry had reviewed and registered this trial (ISRCTN13311651) on 11th March 2024. Any changes or modifications to the protocol before or during the study data collection will be informed to the ethics committee, research team, and the participants if needed.

The participants' recruitment will be initiated in October 2023 and is scheduled to be terminated in June 2024. Then, evaluation and data analysis of the study outcomes will be performed from July to September 2024.

## Sampling

We will recruit participants in the DNCC's One Day Dare Unit and inpatient units. The eligibility criteria will be (1) female aged 18 years or older, (2) being diagnosed with gynecologic cancer (including cervical, ovarian, uterine, vulvar, vaginal, and fallopian tube cancer), (3) receiving chemotherapy at the DNCC, (4) having a smartphone, (5) willing and physically or cognitively able to participate in the study and follow the study procedures, and (6) being able to communicate in Bahasa Indonesia.

On the other hand, patients diagnosed with severe neurological conditions such as unmanaged mental health diagnosis, current metastases to the brain, delirium, and dementia, as well as those with any prior history of receiving chemotherapy or cancer recurrence, will be excluded because these conditions might confound cognitive functioning [24]. Patients with severe hearing or visual impairment that would prevent them from communicating and using the educational digital material will also be excluded. Another exclusion criterion is patients who are single, divorced, or widowed with no sexual partner.

Thirty participants will be recruited using a convenience sampling technique to enroll in the study. A sample size of 30 participants is considered sufficient for evaluating the feasibility of a clinical trial and providing preliminary estimates of the effect sizes of the study outcomes [24]. Before the data collection, the participants will be provided with an explanation of the

study activities and their rights, as detailed in the information sheet. The participants will be able to ask questions about any unclear aspects to the researchers and to participate in this research voluntarily. Subsequently, if a participant agrees to be involved in this study, she will sign the Informed Consent Form prepared by the study team. Meanwhile, the participant can withdraw from the study at any time.

## Outcomes measures

**Primary outcomes.** The primary outcomes will be assessed at the end of the study period through quantitative and qualitative data collection. The data will cover participant recruitment and retention, the implementation of the intervention, and participant satisfaction.

The research team will maintain a log book to record: (1) recruitment rate: the number of gynecological cancer patients approached to participate in the study, (2) response rate: the number of patients who met the eligibility criteria and signed informed consent, (3) participation rate: the number of participant attendances in social media-based education and tele-coaching sessions, (4) facilitators' satisfaction rate, (5) participants' satisfaction rate, and (6) attrition rate: the number of participants who received the intervention and were retained in the study until the final follow-up (week 10).

We will carry out a qualitative evaluation to assess (1) intervention adherence, (2) acceptability of the recruitment processes, intervention design and delivery, and outcome measurements, (3) barriers to recruitment and intervention delivery, and (4) participant satisfaction. Qualitative data will be collected through semi-structured interviews with the participants after the study's completion. An experienced qualitative researcher will conduct the interviews which will be recorded using an audiotape and transcribed by a research assistant. The qualitative data collection will be terminated when the data reaches saturation.

**Secondary outcomes.** *Sociodemographic and clinical variables*. We will use a questionnaire to collect participants' sociodemographic data (i.e., age, education, marital status, employment, ethnicity, and religious affiliation). We will also obtain clinical data at baseline from the participant's medical record, including gynecological cancer type and stage and cancer treatment details.

*Distress*. Distress is defined by the National Comprehensive Cancer Network (NCCN) as a multifactorial unpleasant emotional experience of a psychosocial, and or spiritual nature that may interfere with the ability to cope effectively with cancer, its physical symptoms, and its treatment [25]. The NCCN recommended that distress be recognized, monitored, and managed sufficiently and timely [25]. The NCCN Distress Management Panel also developed the Distress Thermometer (DT), which originated from Roth's study [26], for the initial, middle, and final distress screening. The DT is a single-item rating scale from 0 (no distress) to 10 (extreme distress) [25]. Participants will be asked to choose the number that best describes the distress level they have experienced over the past week, which could be related or unrelated to cancer. Meta-analyses of studies conducted in Asia and non-English-speaking countries suggested that the optimal cut-off point of DT was four [27, 28]. Therefore, participants who score less than four on the DT are noted to experience mild distress, whereas a distress level of four or above indicates a clinically significant distress level [25]. The DT has been adapted and validated by plenty of studies assessing patients with different cancer types in various languages and countries [25], including Indonesia [29]. The Bahasa Indonesia version of DT, which the NCCN has also verified, will be used in this study to measure the distress level of the participants at baseline and follow-up at weeks 6 and 10.

*Knowledge and practice regarding chemotherapy side effects*. At baseline and follow-up at weeks 6 and 10, the participants will be asked to indicate their knowledge about chemotherapy

side effects, how they manage them, and whether they experienced those symptoms in the past week. These outcomes will be measured using the chemotherapy side effect questionnaire outlined by Almohammadi et al. [30]. The questionnaire consists of two subsets. The first part assesses participants' knowledge of 16 common side effects of chemotherapy, such as nausea and vomiting, fatigue, diarrhea and constipation, mouth ulcers and sores, hair loss, and sexual issues [30]. Yes/no responses will be used in this part of the questionnaire. The second part has seven items to examine the practice of side effect management, for example, "I measure my body temperature" and "I avoid eating spicy or fatty foods." We create four-scale Likert responses (never, seldom, often, and always) in this practice questionnaire. In our prior work, these side effect-related questionnaires have demonstrated their validity and reliability (Cronbach's alpha of 0.720 and 0.708, respectively).

*Self-efficacy.* Self-efficacy, posited by Bandura as a strong predictor of behavior, is conceptualized as the perceived capability to perform specific actions needed to achieve particular goals [31, 32]. Numerous studies have shown the role of self-efficacy in improving self-management among patients with chronic diseases, including cancer [33, 34].

The self-efficacy for managing chronic disease 6-item scale (SEMCD6) initially developed by the Stanford Patient Education Research Center will be used to measure the participants' self-efficacy at baseline, week 6, and week 10. This questionnaire is brief and easy to administer as it only contains six items with a Likert scale ranging from 1 (not at all confident) to 10 (totally confident) to indicate how confident a patient is in managing the disease symptoms [35]. The scale's total score is the mean score over at least four of the six items. A higher score indicates a higher level of self-efficacy. The original version of this SEMCD6 was proven valid and reliable in patients with chronic diseases [35]. Previous studies have shown satisfactory psychometric properties of the several versions of SEMCD6, e.g., Spanish, German, Swedish, and Portuguese [36–38]. This study has assessed the validity and reliability of the Bahasa Indonesia version of SEMCD6. The Cronbach's alpha value of the SEMCD6 Bahasa Indonesia version is 0.813.

*Sexual quality of life (SQoL-F).* Sexual quality of life is the primary outcome of our future RCT and will be measured at baseline, week 6, and week 10 using the Sexual Quality of Life-Female (SQOL-F). SQOL-F was initially designed by Symonds, et al. using data sets from women's health surveys in the United Kingdom and the United States to examine how female sexual dysfunction impacts women's sexual quality of life [39]. This tool comprises 18 questions that capture sexual self-esteem, emotional well-being, and relationship issues in women. Each item is rated on a six-point Likert scale ("completely agree" to "completely disagree"). Questions 1, 5, 9, 13, and 18 have reversed scoring. Therefore, the total score ranges between 18 and 108. A higher score indicates a higher quality of sexual life [39].

SQOL-F has been translated into some languages, for example, Iranian [40], Portuguese [41], Turkish [42], and Indonesia [43]. In the prior work of this study, the Bahasa Indonesia version of SQOL-F is valid and reliable (Cronbach's Alpha = 0.751) among Indonesian gynecological cancer survivors.

## FoRSHE-X intervention

This study intervention consists of two phases of telehealth sessions. Phase I will last five weeks, while phase II will take five weeks. The SPIRIT (Standard Protocols Items: Recommendations for Interventional Trials) 2013 template [44] was used to organize the study's schedule of enrolment, interventions, and assessment as summarized in Fig 1.

**Social media-based education.** We developed digital educational media, videos, and infographics, based on clinical guidelines [45] to equip patients with valuable insights and

| | Enrolment | Allocation | Post-allocation | | | | |
|---|---|---|---|---|---|---|---|
| **TIMEPOINT** | **-$w_1{}^a$** | **0** | **$w_1$** | **$w_{2-5}$** | **$w_6$** | **$w_{7-9}$** | **$w_{10}$** |
| **ENROLMENT:** | | | | | | | |
| Eligibility screen | X | | | | | | |
| Informed consent | X | | | | | | |
| Allocation | | X | | | | | |
| **INTERVENTIONS:** | | | | | | | |
| _Phase 1:_ | | | | | | | |
| Social media-based education | | | X | X | X | X | |
| Telecoaching (compulsory) | | | | X | | | |
| _Phase 2:_ | | | | | | | |
| Telecoaching (on demand) | | | | | X | X | X |
| **ASSESSMENTS:** | | | | | | | |
| _Primary Outcomes_ | | | | | | | |
| Recruitment rate | | | | | | | X |
| Response rate | | | | | | | X |
| Participation rate | | | | | | | X |
| Facilitators' satisfaction rate | | | | | | | X |
| Attrition rate | | | | | | | X |
| Intervention adherence | | | | | | | X |
| Acceptability | | | | | | | X |
| _Secondary Outcomes_ | | | | | | | |
| Sociodemographic data | | X | | | | | |
| Distress | | X | | | X | | X |
| Knowledge and practice regarding chemotherapy side effects | | X | | | X | | X |
| Self-efficacy | | X | | | X | | X |
| Sexual quality of life | | X | | | X | | X |

$^a$ $w$ = _week_

**Fig 1. SPIRIT schedule of enrolment, interventions, and assessments.**

strategies for effectively managing the physical, psychological, and sexual issues that may arise during their treatment journey. The educational contents consist of health management on distress, pain, fatigue, anemia, risk of falling, hair fall, skin problems, sleep problems, difficulty breathing, bleeding, dehydration, constipation, diarrhea, loss of taste and smell, nausea and vomiting, weight loss, edema, sexual issues, and infertility. Our educational materials have been crafted to be visually engaging, featuring eye-catching graphics paired with clear and concise descriptions in Bahasa Indonesia to ensure accessibility and understanding for all patients. Recognizing the prevalence and accessibility of social media, we have chosen to leverage the power of Instagram, one of Indonesia's most widely used social media platforms, as the vehicle through which we provide this patient education.

**Telecoaching.** Telecoaching, a purposeful and patient-centered intervention carried out by certified nurse coaches using Zoom meetings or WhatsApp video calls, is organized around a ten-step procedure designed to empower and support the participants in applying the knowledge they have obtained from our social media-based education to manage the arising problems related to cancer and chemotherapy. The ten steps are (1) building trust and setting the agenda, (2) finding the inner drive, (3) addressing hidden issues, (4) clarifying orientation, (5)

defining goals, (6) identifying options, (7) managing barriers, (8) getting support, (9) taking action, and (10) getting feedback [46].

The first phase lays a solid foundation for the coaching partnership by establishing trust and planning the agenda. The participants' inner motivation must then be tapped in step two to ignite their desire for proactive self-management. The subsequent step will address concerns that may be hidden and obstruct growth. Step four, clarifying orientation, is crucial in coordinating the patient's focus with their health objectives. The joint definition of specific objectives and selecting workable solutions to meet those objectives make up steps five and six. In steps seven and eight, the participants will be coached to develop strategies for overcoming obstacles and exploring avenues of external support. Step nine guides patients to take decisive actions after establishing goals and plans. Lastly, receiving feedback in step 10, which brings the coaching process full circle, guarantees that it is dynamic, flexible, and highly responsive to the changing needs and achievements of the individual [46].

## Data analyses

Descriptive statistics of participants' clinical and demographic characteristics and all study outcomes will be reported. The recruitment and retention rates will be computed to determine the feasibility of a full RCT. A future complete RCT will be deemed feasible if the recruitment rate is at least 50% and the retention rate is at least 50%. Response rates to questionnaires, follow-up rates, and intervention adherence rates will also be calculated to provide further evidence for the feasibility of a definitive RCT. For the secondary objective, we will compare the longitudinal quantitative outcomes at three-time points (before, at the middle, and after the interventions) using the paired t-test (for continuous variables) and McNemar's test (for categorical variables), if the data are normally distributed. The non-normally distributed data will be checked using the Wilcoxon test.

Qualitative data will be analyzed using content analysis by the research team's qualitative researchers. The team will develop a codebook based on the study objective and framework. The predetermined codes include acceptability, implementation, facilitators, barriers, and satisfaction. A qualitative researcher will code the transcripts once the final codes have been defined and agreed upon. Emerging codes can be added to the codebook during the analysis. After the analysis, a detailed description of the findings will be reported. The qualitative findings will inform the acceptability of the study and the strategies or best practices of the study protocol. All research data will be stored in a non-publicly open repository which can be accessed by the principal researcher only to protect participant confidentiality.

The core team for this research consists of nurses with expertise in cancer care. The principal investigator is a professor of nursing who will oversee the study and deliver the telecoaching intervention. The research team members include nurses working in academia and at the DNCC. The nurses at DNCC will help provide a list of patients who may be eligible. In addition, a medical oncology consultant will also be involved in the study implementation process.

## Compensation

Participants will receive an internet data plan worth IDR 100,000 per month for the 2.5-month of the research activities. Additionally, a compensation of IDR 100,000 will be provided to each participant at the end of the FoRSHE-X digital intervention to offset the time spent in this study. Furthermore, the participants who will also be interviewed after the provision of digital education will receive IDR 100,000 each as compensation for transportation and time used during the interview process.

## Discussion

This study will examine whether a definitive trial to evaluate the potential benefits of the FoRSHE-X intervention that combines digital education and nurse-led telecoaching is viable and how it should proceed. Feasibility comprises the practicability of implementing the trial, including participant recruitment and retention, intervention delivery, and data collection. This study will also seek to determine appropriate and acceptable outcome measures to be included in the full RCT. As such, the intervention will be given to everyone who participated in this feasibility and pilot study. Nevertheless, using this approach, we cannot test the randomization strategy for the subsequent RCT.

The FoRSHE-X digital intervention is expected to enhance the self-efficacy of gynecologic cancer survivors in preventing and addressing communication obstacles that may hinder them from continuing therapy and preparing for the post-treatment period. This intervention can connect patients inquiring about information and psychological support from healthcare teams without space and time constraints. Digital technology is believed to expand the healthcare coverage and enhance patient participation in their sustainable self-care. The FoRSHE-X digital intervention has the potential to meet the supportive care needs of gynecologic cancer survivors undergoing chemotherapy. Therefore, the final findings of this pilot are planned to be published and disseminated to the participants, healthcare professionals, and other relevant groups. Anonymous data sets will be included in the publication.

Exploring the participants' perspectives and experiences will also provide valuable insights into the acceptability of the study among cancer survivors who seek care. We will assess how well the FoRSHE-X intervention was received and whether participants found it suitable and beneficial. Potential challenges, uncertainties, and strategies to improve them will also be explored by the participants, including those who drop out of the study. The qualitative part of the study can also help identify other potential impacts of the intervention and its interaction with the context in which it is implemented.

Study participants being recruited only from one hospital may limit the generalizability of the results. In this situation, the research findings will likely predominantly reflect the characteristics and experiences of the individuals involved in the study. As a result, the findings might not be readily applicable to a population that is more diverse or geographically separated, such as people who reside in rural areas with little access to online resources. Rural communities often face particular difficulties in accessing healthcare services and digital technology, which may substantially impact how they respond to telehealth like the one under study. Consequently, while the results may be insightful for the hospital's patient population, they should be considered relevant to a larger, more diverse population, especially those with limited internet access and lower educational levels.

One of the primary outcomes will be the sexual quality of life. Using telecoaching to improve the sexual quality of life for participants and their spouses can be a promising approach, although it may come with several concerns. Firstly, discussions about private matters may give rise to privacy issues, limiting participants' openness and willingness to engage fully in telecoaching sessions. Secondly, participants in coaching programs often encounter various challenges in implementing the coaching recommendations effectively.

One potential major challenge is the resistance to change, as people tend to be more comfortable with their existing mindset and habits. Skills and patience may be needed to overcome such resistance. Thirdly, environmental or resource constraints can be another challenge for this study. Particularly for those living in suburban or rural areas, technological obstacles like poor internet connectivity may hinder effective and smooth communication during sessions. We will explore the participants' perspectives to address the potential challenges of the trial.

This study is expected to improve healthcare access for cancer patients; so, they can complete their treatment. Following this study, the survivors are allowed to gain knowledge for self-care in solving chemotherapy side effects issues and psychological uncertainty through digital companionship anywhere. If the project is successfully supported by the survivors, the utilization of telehealth in increasing health literacy and self-efficacy of cancer patients in Indonesia might be adopted to improve well-ness of other patients with chronic illnesses. This feasibility research will improve web-based chronic disease management in future studies.

## Conclusion

To conclude, our research project is targeted to level up cancer survivors' knowledge and ability to manage their physical and psychological issues during chemotherapy and successfully finalize the therapy. A digital supporting package of education and telecoaching for gynecological cancer survivors will be initially delivered to open the opportunity for these women to receive continuous care in their living environment. The pilot findings will assist in shaping future wider research and practice, with the long-term objective of incorporating telehealth into clinical practices across Indonesia to enhance cancer survivorship.

## Supporting information

**S1 Checklist. SPIRIT 2013 checklist.**
(PDF)

**S1 Protocol. Original protocol, bilingual.**
(PDF)

**S1 Text. Informed consent.**
(PDF)

## Acknowledgments

The authors would like to thank the gynecological cancer participants who will be involved in the pilot of this study and Dharmais National Cancer Hospital clinicians for their invaluable support and guidance in the development and implementation of this research project.

## Author Contributions

**Conceptualization:** Yati Afiyanti, Dyah Juliastuti, Ariesta Milanti, Lina Anisa Nasution, Aprilia Dian Prawesti.

**Formal analysis:** Lina Anisa Nasution.

**Funding acquisition:** Yati Afiyanti.

**Investigation:** Lina Anisa Nasution, Aprilia Dian Prawesti.

**Methodology:** Dyah Juliastuti, Winnie Kwok Wei So.

**Project administration:** Aprilia Dian Prawesti.

**Resources:** Aprilia Dian Prawesti.

**Supervision:** Yati Afiyanti, Dyah Juliastuti.

**Writing – original draft:** Yati Afiyanti, Dyah Juliastuti, Ariesta Milanti.

**Writing – review & editing:** Yati Afiyanti, Dyah Juliastuti, Winnie Kwok Wei So, Ariesta Milanti, Lina Anisa Nasution, Aprilia Dian Prawesti.

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
