## [Decision Letter · Decision Letter 0]

10 Sep 2024

PONE-D-24-05651FoRSHE-X digital health intervention to improve the quality of life during chemotherapy among gynecological cancer survivors in Indonesia: A protocol for a pilot and feasibility studyPLOS ONE

Dear Dr. Juliastuti,

Thank you for submitting your manuscript to PLOS ONE. After careful consideration, we feel that it has merit but does not fully meet PLOS ONE’s publication criteria as it currently stands. Therefore, we invite you to submit a revised version of the manuscript that addresses the points raised during the review process.

**Dear Authors**We receive the required number of reviews and announce our decision on your manuscript.**Our decision is: Minor Comments**

We look forward to receiving your revised manuscript.

Kind regards,

Morteza Arab-Zozani, Ph. D.

Academic Editor

PLOS ONE

Journal Requirements:

2. Thank you for stating the following financial disclosure: This work was supported by the Hibah Publikasi Terindeks International (PUTI) Q1 from the Universitas Indonesia (Grant number: NKB-343/UN2.RST/HKP.05.00/2023) received by YA.   

Reviewers' comments:

Reviewer's Responses to Questions

**Comments to the Author**

1. Does the manuscript provide a valid rationale for the proposed study, with clearly identified and justified research questions?

Reviewer #1: Yes

Reviewer #2: Yes

2. Is the protocol technically sound and planned in a manner that will lead to a meaningful outcome and allow testing the stated hypotheses?

Reviewer #1: Yes

Reviewer #2: Yes

3. Is the methodology feasible and described in sufficient detail to allow the work to be replicable?

Reviewer #1: No

Reviewer #2: Yes

4. Have the authors described where all data underlying the findings will be made available when the study is complete?

Reviewer #1: Yes

Reviewer #2: Yes

5. Is the manuscript presented in an intelligible fashion and written in standard English?

Reviewer #1: Yes

Reviewer #2: No

6. Review Comments to the Author

You may also provide optional suggestions and comments to authors that they might find helpful in planning their study.

Reviewer #1: The article with the title " FoRSHE-X digital health intervention to improve the quality of life during chemotherapy among gynecological cancer survivors in Indonesia: A protocol for a pilot and feasibility study" is a very important and popular topic in the world. Although the article is well be written, it is recommended that the following points consider:

1. Grammatically, revise the all the text of manuscript.

2. Implication outcomes of study is not been mentioned.

3. Strengths and weaknesses of the study, in addition to limitation of implementation, are not been mentioned.

4. The magazine writing format is not been followed.

5. The managerial and political implications of the study are not stated.

Reviewer #2: This manuscript addresses novel approach of using digital intervention for increasing awareness among survivors with gynecological cancers. It represents clear approach and method; however, there are some suggestions that preferably to be considered.

1. Please add some details about any digital-related technique that had been used in other cancer studies and elaborate on its favorable outcome.

2. Add conclusion that includes a summary for this study aim, possibly achieved goals, and limitations.

3. The manuscript needs to be well-revised for language grammar and punctuations. Please, confirm it.

7. PLOS authors have the option to publish the peer review history of their article (what does this mean?). If published, this will include your full peer review and any attached files.

Reviewer #1: No

Reviewer #2: No

---

## [Author Response · Author response to Decision Letter 0]

19 Sep 2024

Dear Editors,

We have tried to respond to all your comments, as can be seen in the attached response table. 

1. manuscript has been revised according to the PLOS ONE template

2.“The funders had no role in study design, data collection and analysis, decision to publish, or preparation of the manuscript” statement was added on the manuscript.

3. A data availability statement has been added.

4. Incomplete references have been corrected, and retracted ones have been replaced with relevant current papers.

5. The change on the funding statement was also added to the new cover letter.

6. The grammar has been revised thoroughly. Proofread has been done. 

7. Possible strengths, weaknesses, and study implications have been mentioned and rearranged in the discussion section. 

8. Conclusion has been added.

I hope they meet your expectation. Thank you for advising our manuscript. 

Best regards, 

Dyah

---

## [Decision Letter · Decision Letter 1]

22 Oct 2024

FoRSHE-X digital health intervention to improve the quality of life during chemotherapy among gynecological cancer survivors in Indonesia: A protocol for a pilot and feasibility study

PONE-D-24-05651R1

Dear Dr. Juliastuti,

We’re pleased to inform you that your manuscript has been judged scientifically suitable for publication and will be formally accepted for publication once it meets all outstanding technical requirements.

Kind regards,

Morteza Arab-Zozani, Ph. D.

Academic Editor

PLOS ONE

Additional Editor Comments (optional):

Reviewers' comments:

Reviewer's Responses to Questions

**Comments to the Author**

1. Does the manuscript provide a valid rationale for the proposed study, with clearly identified and justified research questions?

Reviewer #2: Yes

2. Is the protocol technically sound and planned in a manner that will lead to a meaningful outcome and allow testing the stated hypotheses?

Reviewer #2: Yes

3. Is the methodology feasible and described in sufficient detail to allow the work to be replicable?

Reviewer #2: Yes

4. Have the authors described where all data underlying the findings will be made available when the study is complete?

Reviewer #2: Yes

5. Is the manuscript presented in an intelligible fashion and written in standard English?

Reviewer #2: Yes

6. Review Comments to the Author

You may also provide optional suggestions and comments to authors that they might find helpful in planning their study.

Reviewer #2: Comments are addressed. Manuscripts looks better and suitable now for acceptance. Also, Please, double confirm that all grammatical issues and typos are cleared.

7. PLOS authors have the option to publish the peer review history of their article (what does this mean?). If published, this will include your full peer review and any attached files.

Reviewer #2: No

---

## [Editor Report · Acceptance letter]

3 Dec 2024

PONE-D-24-05651R1 

PLOS ONE

Dear Dr. Juliastuti, 

I'm pleased to inform you that your manuscript has been deemed suitable for publication in PLOS ONE. Congratulations! Your manuscript is now being handed over to our production team.

Kind regards, 

on behalf of

Dr. Morteza Arab-Zozani 

Academic Editor

PLOS ONE